# HEATED-UP SOFTMAX EMBEDDING

## ABSTRACT

Metric learning aims at learning a distance which is consistent with the semantic meaning of the samples. The problem is generally solved by learning an embedding, such that the samples of the same category are close (compact) while samples from different categories are far away (spread-out) in the embedding space. One popular way of generating such embeddings is to use the second-to-last layer of a deep neural network trained as a classifier with the softmax cross-entropy loss. In this paper, we show that training classifiers with different temperatures of the softmax function lead to different distributions of the embedding space. And finding a balance between the compactness, "spread-out" and the generalization ability of the feature is critical in metric learning. Leveraging these insights, we propose a "heating-up" strategy to train a classifier with increasing temperatures. Extensive experiments show that the proposed method achieves state-of-the-art performances on a variety of metric learning benchmarks.

## 1   INTRODUCTION

Metric learning is a fundamental research topic in machine learning and has been widely explored in a variety of computer vision applications such as clustering (Xing et al., 2003; Ye et al., 2007), image retrieval (Lee et al., 2008; Zhang et al., 2017), face recognition (Guillaumin et al., 2009; Schroff et al., 2015), and person re-identification (Koestinger et al., 2012; Lisanti et al., 2017).

The objective of metric learning is usually formulated (Chopra et al., 2005; Hoffer & Ailon, 2015; Schroff et al., 2015; Zhang et al., 2017) as learning a metric space which is "compact", where samples from the same classes are close to each other, and "spread-out", where two randomly sampled non-matching features are far away to each other. The learned embedding should also be able to generalize well to unseen classes and samples of the same domain.

One solution for this problem is to define a loss function that tries to explicitly enforce the properties of compactness and spread in the metric space. Two of the most popular loss functions are the contrastive loss (Chopra et al., 2005) and the triplet loss (Hoffer & Ailon, 2015). However, such losses face challenges in sampling, as usually there are a very large number of constraints (possible pairs or triplets) in one dataset and most of them become quickly non-informative in the training process.

Using features from the second-to-last (a.k.a. bottleneck) layer of a deep network trained as a classifier with the softmax function and the cross-entropy loss works well for many metric learning based applications (Razavian et al., 2014) such as image retrieval (Babenko & Lempitsky, 2015) and face verification (Liu et al., 2017). However, the goals of classifier training and metric learning are different, i.e. finding the best decision function vs. learning a "compact" and "spread-out" embedding. This motivates us to investigate the relation between metric learning and classifier training.

In this paper, we study the gradient of the embeddings and show how the temperature parameter in the softmax function (defined by Hinton et al. (2015) for knowledge transfer) plays a crucial role in determining the distribution of the embeddings. This motivates a "heating-up" method that learns a classifier with normalized features and weights using first an intermediate temperature and increasing it during training. The proposed "heating-up" strategy produces compact and spread-out embeddings achieving state-of-the-art performance in deep metric learning.

## 2 RELATED WORKS

Siamese networks with contrastive loss (Chopra et al., 2005) was one of the earliest attempts to solve the metric learning problem. By sampling either two data samples from the same category (positive pair) or two different categories (negative pair), contrastive loss tries to pull two points from positive pair together and push away the points from negative pair. Triplet loss (Hoffer & Ailon, 2015) further requires a margin between the positive and negative pairs distances. One of the main issues of these losses is that the number of possible pairs or triplets is extremely large for a large dataset.

A reasonable solution to address the sampling issue is mining samples that are the most informative for training, also known as "hard mining". There is a large body of works addressing this problem (Schroff et al., 2015; Mishchuk et al., 2017; Harwood et al., 2017; Yuan et al., 2017; Wu et al., 2017). Semi-hard mining (Schroff et al., 2015) tries to find triplets in a training batch, for which the distance of the positive pair and the distance of the negative pair are within a certain margin. HardNet (Mishchuk et al., 2017) is designed to mine some of the hardest triplets within one training batch.

Designing structured losses to consider all the possible training pairs or triplets within one training batch and perform "soft" hard mining can be an alternative solution for hard mining (Song et al., 2017; 2016; Ustinova & Lempitsky, 2016). Lifted structured loss (Song et al., 2016) exploits all triplets in a training batch and provides a smooth loss function for hard mining. A few deep clustering based losses (Law et al., 2017; Song et al., 2017) have also been proposed to solve the problem. Proxy NCA (Movshovitz-Attias et al., 2017) proposes to learn semantic proxies for training data and use a NCA loss for training. Applying hard mining with proxies is more efficient than with samples.

In face verification, training a classifier and using the output of the bottleneck layer as embedding performs reasonably well (Wang et al., 2017b). For this task, the normalization of features (Ranjan et al., 2017), weights (Liu et al., 2017) or both (Wang et al., 2017a) have been explored. In order to achieve promising results, a learnable or fixed scalar is required to be multiplied to the final logits in the softmax function. There are preliminary discussions (Wang et al., 2017a; Ranjan et al., 2017) about the influence of this scalar on the embedding.

This paper shows that the scalar can be seen as the temperature parameter of the softmax function in Hinton et al. (2015), where its influence on the logits for the task of knowldege transfer was studied. The temperature was also explored as a post-processing step to calibrate a pre-trained model (Guo et al., 2017) and produce probability estimates aligned with the expected accuracy of the model. Differently from the above works, which focus on the output probability, the goal of this paper is to study the features from the bottleneck layer.

We analyze how the temperature parameter controls the distribution of the *embedding* by assigning different gradients to different samples and weights. Inspired by these findings, we propose a "heating-up" strategy for training the embedding, which uses increasing temperature while training the classifier. The proposed method makes the embedding trained simply with the softmax cross-entropy loss achieve comparable or better performance than state-of-the-art deep metric learning methods.

## 3 THE HEATED-UP EMBEDDING

### 3.1 REVISITING SOFTMAX EMBEDDING WITH TEMPERATURE

Given a set of $n$ labelled training samples $\{(\mathbf{x}_1, y_1), \ldots, (\mathbf{x}_n, y_n)\}$, where $\mathbf{x}_i \in \mathbb{R}^d$ is the feature of the $i$-th sample, $d$ is the number of dimension for the training data, $y_i \in \{1, \ldots, M\}$ is the category label of sample $\mathbf{x}_i$, and $M$ is the number of categories for training samples, we try to learn an embedding function $\mathbf{f}(\cdot) : \mathbb{R}^d \to \mathbb{R}^k$, which maps a data sample to a vector in $\mathbb{R}^k$, such that for all $i, j, p$ with $y_i = y_j \neq y_p$, $l(\mathbf{f}(\mathbf{x}_i), \mathbf{f}(\mathbf{x}_j)) < l(\mathbf{f}(\mathbf{x}_i), \mathbf{f}(\mathbf{x}_p))$, where $l(\cdot, \cdot) : \mathbb{R}^k \times \mathbb{R}^k \to \mathbb{R}$ is a distance function.

We call $\mathbf{f}(\mathbf{x}) \in \mathbb{R}^k$ the embedding of the data sample $\mathbf{x}$ and use $\mathbf{f}$ as $\mathbf{f}(\mathbf{x})$ to simplify the notation. Considering training a linear classifier $\mathbf{W} = [\mathbf{w}_1, \ldots, \mathbf{w}_M] \in \mathbb{R}^{k \times M}$ and $\mathbf{b} = [b_1, \ldots, b_M]^T \in \mathbb{R}^M$, with $\mathbf{w}_m$ and $b_m$ being respectively the weight vector and bias of the $m^{th}$ category classifier, $\mathbf{z} = [z_1, \ldots, z_M]^T = \mathbf{W}^T \mathbf{f} + \mathbf{b} \in \mathbb{R}^M$ is called the logits. The probability that sample $\mathbf{x}$ belongs to category $m \in \{1, \ldots, M\}$ can be predicted by the softmax function as:

$$p(m|\mathbf{x}) = \frac{\exp(z_m/T)}{\sum_{j=1}^{M} \exp(z_j/T)} = \frac{\exp(\alpha z_m)}{\sum_{j=1}^{M} \exp(\alpha z_j)}. \tag{1}$$

$T$, which is normally set to 1, is the temperature as mentioned in Hinton et al. (2015). We set $\alpha = 1/T$ as the reciprocal of the temperature to simplify the notations in the paper.

Assuming the ground-truth distribution of the training sample is $q(m|\mathbf{x})$, generally $q(m|\mathbf{x})$ is a Dirac delta function, which equals 1 if $m = y$ and 0 otherwise, where $y$ is the ground-truth label of $\mathbf{x}$, the cross entropy loss with respect to $\mathbf{x}$, and its gradient with respect to $z_m$ are defined as:

$$\ell(\mathbf{x}, \alpha) = -\sum_{m=1}^{M} \log(p(m|\mathbf{x}, \alpha))q(m|\mathbf{x}) \quad \text{and} \quad \frac{\partial \ell}{\partial z_m} = \alpha(p(m|\mathbf{x}, \alpha) - q(m|\mathbf{x})). \tag{2}$$

Considering $z_m = \mathbf{w}_m^T \mathbf{f} + b_m$, the gradient with respect to the feature $\mathbf{f}$ is:

$$\frac{\partial \ell}{\partial \mathbf{f}} = \sum_{m=1}^{M} \frac{\partial \ell}{\partial z_m} \frac{\partial z_m}{\partial \mathbf{f}} = \alpha \sum_{m=1}^{M} (p(m|\mathbf{x}, \alpha) - q(m|\mathbf{x}))\mathbf{w}_m. \tag{3}$$

And the gradient with respect to the weight $\mathbf{w}_m$ is:

$$\frac{\partial \ell}{\partial \mathbf{w}_m} = \frac{\partial \ell}{\partial z_m} \frac{\partial z_m}{\partial \mathbf{w}_m} = \alpha(p(m|\mathbf{x}, \alpha) - q(m|\mathbf{x}))\mathbf{f}. \tag{4}$$

The following part of this section (Sec. 3.2) will show how $\alpha$ changes the magnitude of the gradient assignment to different features and weights. To show this, we $\ell_2$-normalize the classifier weights $\mathbf{w}_m$ and the feature $\mathbf{f}$ (as in Wang et al. (2017a)). It helps 1) separate the impact of $\alpha$ from the norm of the embedding and the norm of the classifier weights to the softmax function; 2) achieve better performance than the unnormalized model (Wang et al., 2017a) (a detailed comparison between the normalized model and unnormalized model will be given in Sec. 4). Sec. 3.3 will study the effect of the normalization . Exploiting these findings to derive an effective strategy to learn compact and spread-out embeddings, we finally detail our proposed "heating-up" idea in Sec. 3.4.

## 3.2 Gradient Assignment by $\alpha$

In this subsection, we will show how training a deep classification network with different $\alpha$ values affects the gradients of different training samples and class weights. From Eq. (1), $p(m|\mathbf{x}, \alpha)$ satisfies:

$$\lim_{\alpha \to +\infty} p(m|\mathbf{x}, \alpha) = \begin{cases} 1/K & z_m = \max(z_1, ..., z_M) \\ 0 & \text{otherwise} \end{cases}, \text{ and } \lim_{\alpha \to 0} p(m|\mathbf{x}, \alpha) = 1/M \tag{5}$$

where $K$ is the number of logits whose value equals the maximum logits value. In other words, as $\alpha$ increases, the predicted probability will become more "spiky" at the logits that have the largest value. On the other hand, if $\alpha$ approaches 0, the predicted probability will approach the uniform distribution.

The training samples can be divided into: (i) "incorrect" samples: samples that are not classified as the correct category ($\{\mathbf{x} : \exists m \neq y, z_m \geq z_y\}$); (ii) "correct" samples: samples being correctly classified by the classifier ($\{\mathbf{x} : \forall m \neq y, z_y > z_m\}$). We further define two subtypes of samples in the "correct" category: "boundary" samples are samples close to the decision boundary; "centroid" samples are samples laying close to the center of the region that belongs to the category.

We denote the normalized weights and features as $\hat{\mathbf{w}}_m$ and $\hat{\mathbf{f}}$ respectively. The gradient of the loss w.r.t. the normalized embedding and the normalized weight (i.e. Eq. (3)-(4) with $\hat{\mathbf{f}}$ instead of $\mathbf{f}$ and $\hat{\mathbf{w}}_m$ instead of $\mathbf{w}_m$) exhibits three important properties for gradient assignment for different $\alpha$ values.

**Property 1.** *The gradient of the cross entropy loss with respect to the feature and the weights satisfy:*

$$\lim_{\alpha \to 0} ||\frac{\partial \ell}{\partial \hat{\mathbf{f}}}||_2 = 0, \text{ and } \lim_{\alpha \to 0} ||\frac{\partial \ell}{\partial \hat{\mathbf{w}}_m}||_2 = 0. \tag{6}$$

It's because $-1 \leq p(m|\mathbf{x}, \alpha) - q(m|\mathbf{x}) \leq 1$. Therefore, with very small $\alpha$ (e.g. to the left of the green dashed line in Fig. 1, close to 0), all the samples will get small gradients. Similarly, all the weights will get small gradients.

**Property 2.** *If $\hat{\mathbf{f}}$ is the normalized feature of a "correct" sample, the gradient of the cross entropy loss with respect to the feature satisfies:*

$$\lim_{\alpha \to +\infty} ||\frac{\partial \ell}{\partial \hat{\mathbf{f}}}||_2 = 0. \tag{7}$$

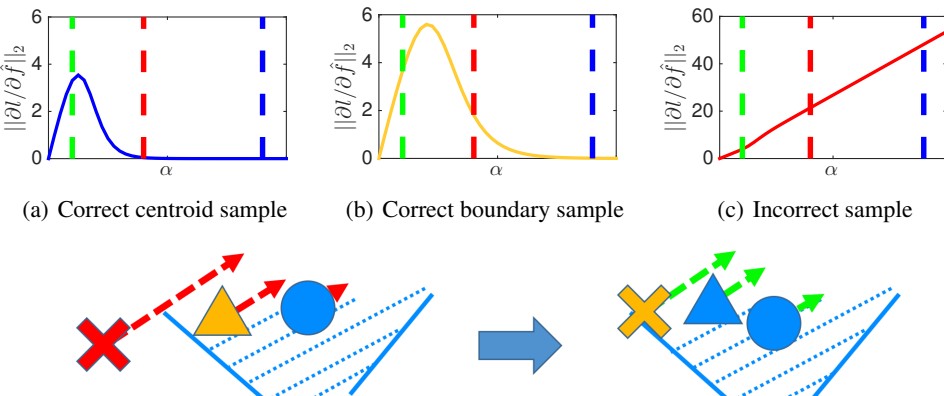

(a) Correct centroid sample     (b) Correct boundary sample     (c) Incorrect sample

(d) Illustration of gradient assignment by the proposed "heating-up" strategy for different samples. The left-hand side shows the first step with intermediate $\alpha$, while the right-hand side shows the second step with smaller $\alpha$.

Figure 1: (a-c) Relation between $\alpha$ and the magnitude of the gradient with respect to the embeddings of "correct" (centroid and boundary) and "incorrect" samples. The gradients are sampled from models trained on the Car196 dataset (Krause et al., 2013). (d) Illustration of our 'heating-up' strategy.

*If $\hat{\mathbf{f}}$ is the normalized feature of an "incorrect" sample, the gradient, in most cases[1], satisfies:*

$$\lim_{\alpha \to +\infty} ||\frac{\partial \ell}{\partial \hat{\mathbf{f}}}||_2 = +\infty \tag{8}$$

We give the detailed proof of property 2 in the Appendix A.1. Overall, with large $\alpha$ (e.g. the blue dashed line in Fig. 1) the magnitudes of the gradients for "incorrect" samples will become very large, while the magnitudes of gradients for "correct" samples will become very small.

**Property 3.** *If $\hat{\mathbf{f}}$ is the normalized feature of a "correct" sample, the gradient of the cross entropy loss with respect to the weight satisfies:*

$$\lim_{\alpha \to +\infty} ||\frac{\partial \ell}{\partial \hat{\mathbf{w}}_m}||_2 = 0. \tag{9}$$

*If $\hat{\mathbf{f}}$ is the normalized feature of an "incorrect" sample, the gradient satisfies:*

$$\lim_{\alpha \to +\infty} ||\frac{\partial \ell}{\partial \hat{\mathbf{w}}_m}||_2 = \begin{cases} +\infty & z_m = \max(z_1, ..., z_M) \text{ or } m = y; \\ 0 & otherwise. \end{cases} \tag{10}$$

A detailed proof will be given in Appendix. A.2. Overall, with large $\alpha$, the network will only consider few weights whose predicted probabilities violate the ground-truth distribution ($\lim_{\alpha \to +\infty} p(m|\mathbf{x}, \alpha) \neq q(m|\mathbf{x})$). We named those weights as "hard" weights.

### 3.3 INFLUENCE OF THE NORMALIZATION

We here discuss the influence of the normalization on the gradient of the loss w.r.t. the embedding. The Jacobian matrix of the $\ell_2$-normalized embedding $\hat{\mathbf{f}}$ with respect to the original embedding $\mathbf{f}$ is:

$$\mathbf{J}_{\hat{\mathbf{f}}}(\mathbf{f}) = \frac{1}{||\mathbf{f}||_2}(\mathbf{I} - \hat{\mathbf{f}}\hat{\mathbf{f}}^T), \tag{11}$$

where $\mathbf{I}$ is the identity matrix. Considering Eq. (3) and the chain rule, we have:

$$\frac{\partial \ell}{\partial \mathbf{f}} = (\frac{\partial \ell}{\partial \hat{\mathbf{f}}})^T \mathbf{J}_{\hat{\mathbf{f}}}(\mathbf{f}) \tag{12}$$

---

[1]The equation may not be satisfied in some special cases. For example, if one positive weight and one negative weight have exactly the same magnitude and direction, then the gradient would be $\mathbf{0}$.

Considering the norm $||\mathbf{f}||_2$ in the denominator, the magnitude of the gradient is inversely proportional to the norm of the embedding. Therefore, even if the normalized embeddings are the same, the gradients w.r.t. the embeddings would still different for embeddings with different norms. Specifically, the embedding with larger norm will have smaller gradient. This may seem as a problem, and one possible solution is to remove the norm term $||\mathbf{f}||_2$ in the denominator. We tried this idea for the experiments in Sec. 5, but it did not give significant improvement. The reason for that is likely that since $\partial\ell/\partial\mathbf{f}$ and $\mathbf{f}$ are always orthogonal (Wang et al., 2017a), updating the feature along the direction of gradient cannot change much the norm of the feature. We did observe that in practice. When applying $\ell_2$-normalization to the feature during training, the norms of features before normalization are very similar. On the contrary, when training without normalization, the norm of the features may have large variations as it is the case for off-the-shelf classifier as detailed in Sec. 4. For numerical stability and ease of implementation we use Eq. (12) to calculate the gradient, but the gradient analysis in Sec. 3.2 still holds for unnormalized features before normalization.

We found out that batch normalization (Ioffe & Szegedy, 2015) without the learned scale[2], $\hat{BN}(\cdot)$, can work slightly better than $\ell_2$-normalization. We define the batch normalized embedding as:

$$\hat{\mathbf{f}}_{BN} = \hat{BN}(\mathbf{f})/\sqrt{k} \tag{13}$$

where $k$ is the number of dimensions of $\mathbf{f}$. Batch normalization tries to make each dimension of the embedding have zero mean and unit variance. Therefore, after batch normalization, the norm of the embedding is roughly $\sqrt{k}$, and the normalized feature $\hat{\mathbf{f}}_{BN}$ has norm close to 1, which is similar to $\ell_2$ normalization. Batch normalization may work better than $\ell_2$-normalization because in fine-grained recognition problem, many embeddings from different categories can be very similar. Batch normalization removes the mean and scales the embedding thus creating more variance. For classifier weights, $\ell_2$-normalization always gives us promising result.

### 3.4 THE "HEATING-UP" STRATEGY

In order to get an intra-category compact, inter-category "spread-out" and well generalized feature, we propose a two-step "heating-up" strategy, which 1) uses an intermediate $\alpha$ (temperature) to start training, and then 2) increases the "temperature" (decreasing $\alpha$ value) and decreases the learning rate.

At the beginning of the training, the network should focus more on the "incorrect" samples and "boundary" samples and quickly moves them to the class center (producing compactness, Fig. 1(d) left). It should also focus more on the "hard" weights, and push them away (producing spread-out). Using an intermediate $\alpha$ for training (e.g. the red dashed line in Fig. 1) satisfies these conditions.

If the training started with a small $\alpha$ (i.e. high temperature), the network would assign similar (and potentially very small, if $\alpha$ is very small) gradients to all samples (left of the green dashed line in Fig. 1(a)-1(c)) and all weights. There are two main issues with this, 1) assigning gradient to the centroid samples at the beginning of the training will move them to a very small region close to the class center. However, in metric learning problem, as the training categories are different from the test categories, this can make the embedding overfit the training categories and not generalize well to test categories; 2) "incorrect" samples and "hard" weights affect the accuracy the most. Failing in giving larger gradients (i.e. higher priority) to those will make the training process inefficient.

On the contrary, choosing a large $\alpha$ (i.e. very low temperature) for the initial training would assign large gradients to "incorrect" samples and very small gradients to all "correct" ("centroid" and "boundary") samples (right of the red dashed line in Fig. 1(a)-1(c)). Since the "boundary" samples will not get enough update, they will stay near the decision boundary. Therefore, the embedding of the samples of the same category will not be compact (as shown in Appendix C, features of the same category are more compact for the model trained with smaller $\alpha$ values). Also, the network will only focus on the "hard" weights and not pushing away other weights, which makes the weights vector not spread-out. The non spread-out weight vectors results in the features not being spread-out, since the features are moved towards corresponding weights.

After the first step of training with an intermediate $\alpha$, most of the "incorrect" samples become "boundary" samples and most of the "boundary" samples become "centroid" samples (Fig. 1(d), right), in order to make the feature more compact, we increase the temperature (decrease $\alpha$ value). It helps

---

[2]in this paper, batch normalization always refers to batch normalization without learned scale

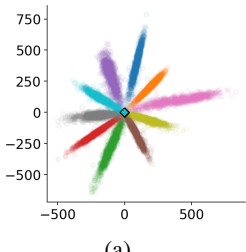 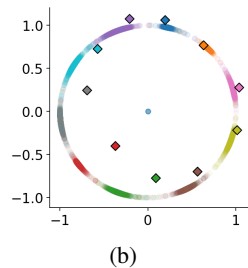 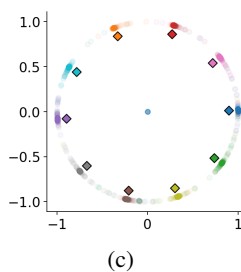

| (a) | (b) | (c) |

Figure 2: 2(a)-2(b): Embedding and normalized embedding of a classifier trained without $\ell_2$ normalization. 2(c): Embedding obtained with a classifier trained with $\ell_2$ normalization and $\alpha = 4$.

assign sufficient gradient to all "correct" samples and push them towards the class center. In practice, we also reduce the learning rate to avoid overfitting.

Multiple strategies could be defined to increase the temperature during training: (i) gently increasing the temperature; (ii) training with the starting temperature until convergence and using a higher temperature to fine-tune the trained network. Both methods lead to similar performance. However, since the former method would introduce an additional parameter to control the speed of increase of the temperature, we used the latter in our experiments in Sec. 5. The visualization of the feature learned with different models on Car196 dataset (Krause et al., 2013) is provided in Appendix C.

## 4 COMPARING WITH OFF-THE-SHELF CLASSIFIER

It is interesting to compare the embedding from an off-the-shelf classifier, i.e. trained with unnormalized features and weights and $\alpha = 1$, with the embeddings obtained by a model trained with normalized features and weight and some fixed $\alpha$ value. We train a LeNet (LeCun et al., 2015) model on the MNIST dataset, and set the number of nodes in the bottleneck layer to 2 for visualization. 50,000 samples are used for training, and 10,000 different test samples are used to draw the figures.

In Fig. 2, different colors represent different digits, and each diamond corresponds to the weight of the classifier of the corresponding digit (For better visualization, the weight are slightly moved towards origin). We can see in Fig.2(a), as observed in other works (Wang et al., 2017a; Ranjan et al., 2017), that for an off-the-shelf classifier (i) the magnitude of the embedding can be extremely large, since "correct" samples with larger norm will produce smaller loss (Wang et al., 2017a); (ii) the embedding is not "compact". Even if the feature is $\ell_2$-normalized (Fig. 2(b)), the embedding is still not as "compact" as the feature trained with normalization and proper $\alpha$ values (Fig. 2(c), see detailed analysis in Appendix B.1).

Training with $\ell_2$-normalization for the feature is different from simply applying normalization to the final feature in the test phase. As discussed and shown in Sec. 3.3 in Eq.(11) and (12), $\ell_2$-normalization will change the gradient of each sample during training.

## 5 EXPERIMENTS ON METRIC LEARNING

We conduct experiment on the following fine-grained datasets, using the training/test splits of Movshovitz-Attias et al. (2017). In all these datasets, the categories in the training and test splits do *not* overlap. For the sake of completeness, we also report fine-grained classification results on Car196 and CUB200 in Appendix D.

- **Cars** (Car196) (Krause et al., 2013) is a fine-grained car category dataset, which contains 16,185 images of 196 car models. 8,054 images of the first 98 categories are used for training, while the 8,131 images of the other 98 categories are used for test.
- **Caltech-UCSD Birds-200-2011** (CUB200) (Welinder et al., 2010) is a fine-grained dataset which contains 11,788 images of 200 bird species. 5,864 images of the first 100 species are used for training, while the 5,924 images of the other 100 species are used for test.
- **Stanford Online Product** (Product) dataset (Song et al., 2016) contains 120,053 images of 22,634 products categories. 59,551 images of 11,318 categories are used for training, while the other 60,502 images from 11,316 categories are kept for test.

- **In-shop Clothes Retrieval** (Fashion) dataset (Liu et al., 2016) contains 54,642 images of 11,735 categories of fine-grained clothes. The dataset is split into 3 subsets. 52,712 images of 7,982 categories are used for training. The other 28,760 images of 3985 categories are kept for test, split into a gallery set (12,612 images) and a query set (14,218 images).

## 5.1 IMPLEMENTATION DETAILS

We use the TensorFlow Deep Learning framework (Abadi et al., 2016) to implement the proposed method. For fair comparison, we exactly follow the details in Movshovitz-Attias et al. (2017). We use a GoogLeNet V1 (Szegedy et al., 2015) network pre-trained on ILSVRC 2012-CLS data (Russakovsky et al., 2015) as base network. The input images are all resized to $256 \times 256$. For training, the resized images are randomly crop to $224 \times 224$ with random horizontal flipping. In test phase, we use one single center crop as in Movshovitz-Attias et al. (2017). For Car196, CUB200 and fashion datasets the network is fine-tuned by SGD optimizer with 0.9 momentum. The learning rate is set to 0.004. For product dataset, the optimizer is ADAM with learning rate of 0.01. The embedding size is set to 64 and the batch size is set to 32. We choose $\alpha = 16$ for all the datasets, as an "intermediate" temperature, it works well for different embedding sizes (see Sec. 5.3). For "heating-up", $\alpha$ will decrease (temperature increases on the other hand) from 16 to 4 and the learning rate will decrease to 1/10 of the original learning rate. The training process usually converges within 50 training epochs, which is similar to the fastest state-of-the-art method, ProxyNCA (Movshovitz-Attias et al., 2017).

## 5.2 EVALUATION

Following previous works on metric learning, we evaluate the clustering quality and the retrieval performance on the images of the test set. All the features are $\ell_2$-normalized before calculating the evaluation metric, as in Song et al. (2017). The normalized features performs slightly better than the unnormalized feature. For clustering, the K-Means algorithm is run on all the embeddings of the test samples. The number of cluster is chosen to be the number of categories in the test set. Each test sample will be assigned a cluster index according to which cluster it belongs to. Normalized Mutual Information (NMI) (Schütze et al., 2008) between the clustering index and the ground-truth label is used as the metric for clustering.For retrieval, the performance is evaluated by Recall@K, which is also a widely used metric for this problem. Given a query sample from the test set, K samples from the rest of the test set (or the gallery set for the fashion dataset) with the smallest distance are retrieved. If any retrieved sample is from the same category as the query sample, the recall for this query is 1, otherwise, 0. The reported Recall@K is the average recall on the whole test set.

We train a classifier on the training dataset with the softmax function and cross-entropy as baseline (SM). For the baseline classifier, in training, the features and the weights are not normalized and $\alpha$ is set to 1. 4 different versions of classifiers trained with the proposed methods are used for evaluation:

- **LN**: softmax with $\ell_2$-normalized embedding, $\ell_2$-normalized weights and $\alpha = 16$.
- **BN**: softmax with batch normalized embedding, $\ell_2$-normalized weights and $\alpha = 16$.
- **HLN**: Heated-up model using $\alpha = 4$ to fine-tune LN.
- **HBN**: Heated-up model using $\alpha = 4$ to fine-tune BN.

We also compare the proposed method with state-of-the-art metric learning methods. Existing literatures are using different base networks and different evaluation protocols. For fair comparison, only the methods using GoogleNetV1 as base network and Euclidean distance as the final evaluation metric are listed: [1] Triplet learning with semi-hard negative mining (Schroff et al., 2015), [2] Lifted structured loss (Song et al., 2016), [3] Learnable Structured Clustering (Song et al., 2017); [4] Deep Clustering Learning without spectral learning (Law et al., 2017); [5] Deep Metric Learning with Smart Mining (Harwood et al., 2017) and [6] ProxyNCA (Movshovitz-Attias et al., 2017). For the Fashion dataset, we compared with [7] FashionNet (Liu et al., 2016) and [8] Hard-Aware Deeply Cascaded Embedding (Yuan et al., 2017).

## 5.3 METRIC LEARNING

The performances of all the methods on all datasets are listed in Tables 1 and 2. The softmax baseline already shows comparable result with many other triplet loss based methods. The embeddings trained with either $\ell_2$-normalization or batch normalization improve the performance of the softmax baseline.

Table 1: NMI and Recall(%) for the Car196, CUB200 and Stanford datasets

|  | [1] | [2] | [3] | [4] | [5] | [6] | SM | LN | BN | HLN | HBN |
|---|---|---|---|---|---|---|---|---|---|---|---|
| CAR196 DATASET | | | | | | | | | | | |
| NMI | 53.35 | 56.88 | 54.44 | 61.12 | 59.50 | 64.90 | 59.52 | 62.40 | 65.81 | 66.87 | **68.10** |
| R@1 | 51.54 | 52.98 | 58.11 | 67.54 | 64.65 | 73.22 | 60.76 | 68.59 | 71.12 | 71.93 | **74.70** |
| R@2 | 63.78 | 66.70 | 70.64 | 77.77 | 76.20 | 82.42 | 73.58 | 78.55 | 80.62 | 81.68 | **83.90** |
| R@4 | 73.52 | 76.01 | 80.27 | 85.74 | 84.23 | 86.36 | 82.50 | 86.18 | 87.82 | 88.34 | **89.77** |
| CUB200 DATASET | | | | | | | | | | | |
| NMI | 55.38 | 56.50 | 59.23 | 56.87 | 59.90 | 59.53 | 57.19 | 59.23 | 59.20 | 60.34 | **60.75** |
| R@1 | 42.59 | 43.57 | 48.18 | 50.08 | 49.78 | 49.21 | 44.02 | 46.86 | 47.27 | 49.68 | **50.68** |
| R@2 | 55.03 | 56.55 | 61.44 | 62.24 | 62.34 | 61.90 | 55.86 | 59.79 | 59.67 | 61.85 | **62.58** |
| R@4 | 66.44 | 68.59 | 71.83 | 73.38 | **74.05** | 67.90 | 68.18 | 71.56 | 71.89 | 73.08 | 73.82 |
| STANFORD PRODUCT DATASET | | | | | | | | | | | |
| NMI | 89.46 | 88.65 | 89.48 | 88.70 | - | **90.60** | 88.66 | 90.11 | 90.45 | 90.39 | **90.61** |
| R@1 | 66.67 | 62.46 | 67.02 | 64.52 | - | **73.70** | 63.94 | 69.51 | 71.19 | 70.36 | 72.04 |
| R@10 | 82.39 | 80.81 | 83.65 | 82.53 | - | - | 80.07 | 84.69 | 85.89 | 85.41 | **86.25** |
| R@100 | 91.85 | 91.93 | 93.23 | 92.35 | - | - | 90.28 | 92.97 | 93.75 | 93.70 | **93.80** |

Table 2: Recall(%) for the In-shop Clothes Retrieval Dataset

|  | [7] | [8] | SM | LN | BN | HLN | HBN |
|---|---|---|---|---|---|---|---|
| R@1 | 53.0 | 62.1 | 78.6 | 79.6 | 80.7 | 80.5 | **81.1** |
| R@10 | 73.0 | 84.9 | 93.7 | 94.2 | **94.4** | 94.2 | 94.2 |
| R@20 | 76.0 | 91.2 | 95.4 | 96.0 | **96.1** | **96.1** | 95.9 |
| R@30 | 77.0 | 92.3 | 96.3 | 96.8 | **96.9** | 96.7 | **96.9** |

Since in test phase, all the features are $\ell_2$ normalized before calculating the metric, the performance gain is not coming from the $\ell_2$-normalization of the final features. Batch normalization works slightly better than $\ell_2$-normalization. The "heated-up" models (HLN and HBN) show better performance in almost all the metrics compared to the embedding trained with a fixed temperature.

We further study how different embedding sizes and $\alpha$ values affect the retrieval performance. The size of the embedding is chosen in $[64, 128, 256]$, and the $\alpha$ value in $[4, 8, 16, 32, 64]$. Normalization is applied to both the feature and the weight. The R@1 metric on the test set with different embedding sizes and $\alpha$ values is reported in Table 3. Performances of the feature learned by softmax function without normalization and feature learned from "heated-up" model are also given as "SM" and "HBN". The "heated-up" model outperforms all the other models by a significant margin. Between models trained with fixed $\alpha$ values, the model using $\alpha = 16$ outperforms others.

Table 3: R@1(%) for Car196 with Different $\alpha$ Values and Embedding Sizes

| MODEL #DIM | SM | $\alpha = 4$ | $\alpha = 8$ | $\alpha = 16$ | $\alpha = 32$ | $\alpha = 64$ | HBN $(16 \rightarrow 4)$ |
|---|---|---|---|---|---|---|---|
| 64 | 60.8 | 67.4 | 68.7 | 71.1 | 69.5 | 62.5 | **74.7** |
| 128 | 65.2 | 71.6 | 71.0 | 74.2 | 73.0 | 66.6 | **77.5** |
| 256 | 67.3 | 72.2 | 69.7 | 78.0 | 75.2 | 70.1 | **80.1** |

# 6 DISCUSSION

We have discussed how the temperature parameter in the softmax function affects the distribution of the embedding in the second last layer of a deep classification model. Training with an intermediate temperature will lead to an intra-category compact and inter-category "spread-out" embedding which is beneficial for both clustering and retrieval. A "heating-up" method is also proposed to further improve the clustering and retrieval performance of the embedding by fine-tuning with a higher temperature. Our classifier based approach achieves good performance in metric learning problems with a simple and efficient training process.

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

APPENDIX

## A  GRADIENT ANALYSIS FOR CLASSIFIER WITH NORMALIZATION AND TEMPERATURE

### A.1  PROOF OF PROPERTY 2

Eq (3) contains $M$ terms in the sum, one term for each of the $M$ categories. There is 3 types of terms: type 1, term with respect to the ground-truth category; type 2, term with respect to the logits that has the largest value and does not belong to the ground-truth category; type 3, other terms.

For $\alpha \rightarrow +\infty$, considering the term of type 1 for "correct" samples first, since $\lim_{\alpha \rightarrow +\infty} p(y|\mathbf{x}, \alpha) = 1$ and $q(y|\mathbf{x}) = 1$, according to the property of exponential function, $\lim_{\alpha \rightarrow +\infty} \alpha(p(y|\mathbf{x}, \alpha) - q(y|\mathbf{x})) = 0$ and the magnitude for this term will approach 0. For other terms, the magnitude will also approach 0, due to $\lim_{\alpha \rightarrow +\infty} p(m|\mathbf{x}, \alpha) = 0$, and $q(m|\mathbf{x}) = 0, m \neq y$. Therefore, the magnitude of the gradient will always approach to 0.

Considering "incorrect" samples, the magnitude of the type 1 term $(\alpha(p(y|\mathbf{x}, \alpha) - q(y|\mathbf{x}))\hat{\mathbf{w}}_y)$ will approach $+\infty$ as $\alpha \rightarrow +\infty$, because $p(y|\mathbf{x}, \alpha)$ will approach either 0 or $1/K$ $(K \geq 2)$[3] and $q(y|\mathbf{x}) = 1$. Similarly, the magnitude of the term of type 2 will also approach $+\infty$. For other terms, due to the property of the exponential function, that if $z_m \neq \max(z_1, ..., z_M)$, $\lim_{\alpha \rightarrow +\infty} \alpha p(m|\mathbf{x}, \alpha) = 0$, the magnitude of any term of type 3 will decrease to 0. Therefore, for "incorrect" samples, as $\alpha \rightarrow +\infty$, unless in some special cases[4], the magnitude of the gradient with respect to the normalized embedding will approach infinity.

### A.2  PROOF OF PROPERTY 3

From Eq. (1) and Eq. (5), similar to the discussion in Appendix A.1, for the normalized feature $\hat{\mathbf{f}}$ of a "correct" sample, $\lim_{\alpha \rightarrow +\infty} \alpha(p(y|\mathbf{x}, \alpha) - 1) = 0$ and $\lim_{\alpha \rightarrow +\infty} \alpha p(m|\mathbf{x}, \alpha) = 0, \forall m \neq y$. Applying these to Eq. (4), Eq. (9) is proved.

For the normalized feature $\hat{\mathbf{f}}$ of an "incorrect" sample, similar to the discussion in Appendix. A.1, when $\lim_{\alpha \rightarrow +\infty} (p(m|\mathbf{x}, \alpha) - q(m|\mathbf{x})) \neq 0$, which means $z_m = \max(z_1, ..., z_M)$ or $m = y$, the magnitude of the gradient will approach to $+\infty$. Otherwise, the magnitude of the gradient will approach to 0.

## B  THE GRADIENT ANALYSIS FOR OFF-THE-SHELF CLASSIFIER

To understand the feature learned with the off-the-shelf classifier, we need to study the gradient of the loss with respect to the feature and the gradient of the loss with respect to the classifier weights.

### B.1  GRADIENT WITH RESPECT TO THE FEATURE

By setting $\alpha = 1$, considering $z_j = ||\mathbf{f}||_2 \mathbf{w}_j^T \hat{\mathbf{f}}$, Eq. (1) becomes:

$$p(m|\mathbf{x}) = \frac{\exp(||\mathbf{f}||_2 \mathbf{w}_m^T \hat{\mathbf{f}})}{\sum_{j=1}^{M} \exp(||\mathbf{f}||_2 \mathbf{w}_j^T \hat{\mathbf{f}})} \tag{14}$$

$p(m|\mathbf{x})$ satisfies:

$$\lim_{||\mathbf{f}||_2 \rightarrow +\infty} p(m|\mathbf{x}) = \begin{cases} 1/K & z_m = \max(z_1, ..., z_M) \\ 0 & \text{otherwise,} \end{cases} \tag{15}$$

where $K$ is the number of logits whose value equals the maximum logits value. In other words, as $||\mathbf{f}||_2$ increases, the predicted probability will be spikier at the logits that have the largest value.

---

[3] $K$ can't be 1, otherwise $\mathbf{x}$ is a "correct" sample.

[4] For example, two terms having exactly the same magnitude but opposite direction.

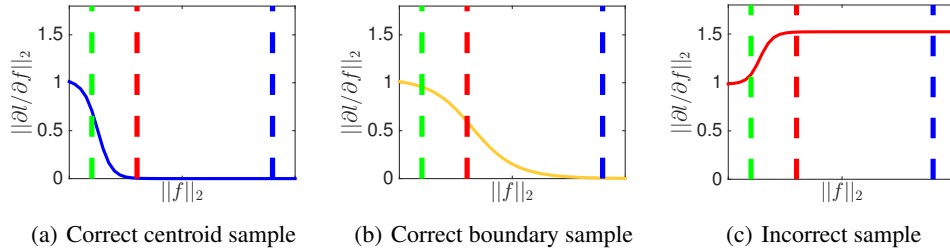

(a) Correct centroid sample      (b) Correct boundary sample      (c) Incorrect sample

Figure 3: Relation between the norm of the feature and the magnitude of the gradient with respect to "correct" (centroid and boundary) and "incorrect" samples for SM model. The green dashed lined correspond to a small feature norm, the red one to an intermediate feature norm and the blue one to a large feature norm. The gradients are sampled from models trained with Car196 dataset.

By setting $\alpha = 1$, Eq. (3) becomes,

$$\frac{\partial \ell}{\partial \mathbf{f}} = \sum_{m=1}^{M} \frac{\partial \ell}{\partial z_m} \frac{\partial z_m}{\partial \mathbf{f}} = \sum_{m=1}^{M} (p(m|\mathbf{x}) - q(m|\mathbf{x})) \mathbf{w}_m \tag{16}$$

We also consider the 3 types of terms in Eq. (16) as defined in Sec. 3.2. When $||\mathbf{f}||_2 \to +\infty$, for "correct" samples, the predicted probability will become a one-hot vector, and all the terms in Eq. 16 will approach to $\mathbf{0}$. Therefore, the overall gradient will approach to $\mathbf{0}$ and the magnitude of the gradient is 0.

For "incorrect" samples, when $||\mathbf{f}||_2 \to +\infty$, since $p(y|\mathbf{x})$ will approach to either 0 or $1/K$ and $q(y|\mathbf{x}) = 1$, type 1 term will become either $-\mathbf{w}_y$ or $(1/K - 1)\mathbf{w}_y$, both are fixed vector. Similarly, all type 2 terms will become fixed vectors $(1/K\mathbf{w}_m)$. Type 3 terms will approach to $\mathbf{0}$. Therefore, Eq. 16 will approach to a fixed vector (the vector sum of type 1 term and type 2 terms) and the magnitude of the gradient will approach to a constant. Unless some special cases, the constant is not 0.

Thus, the properties used in Sec. 4 have been proved. Similarly to Figure 1, the relation between the magnitude of the gradient w.r.t. the norm of the feature for "incorrect" and "correct" samples is shown in Fig. 3. As boundary features trained with large $\alpha$, boundary features (Fig. 3(b)) with large norm (blue dashed line in Fig. 3) will not get enough update thus leading to a not compact embedding.

### B.2 GRADIENT WITH RESPECT TO THE WEIGHTS

Considering the gradient with respect to the weights, from Eq. 2, we have

$$\frac{\partial \ell}{\partial \mathbf{w}_m} = \frac{\partial \ell}{\partial z_m} \frac{\partial z_m}{\partial \mathbf{w}_m} = (p(m|\mathbf{x}) - q(m|\mathbf{x})) \mathbf{f} = ||\mathbf{f}||_2 (p(m|\mathbf{x}) - q(m|\mathbf{x})) \hat{\mathbf{f}} \tag{17}$$

From Eq. (14), Eq. (15) and the property of the exponential function, similar to the discussion in Appendix. A.2, we have, for a "correct" sample feature $\mathbf{f}$, $\lim_{||\mathbf{f}||_2 \to +\infty} ||\mathbf{f}||_2 (p(y|\mathbf{x}) - 1) = 0$ and $\lim_{||\mathbf{f}||_2 \to +\infty} ||\mathbf{f}||_2 p(m|\mathbf{x}) = 0, \forall m \neq y$, and therefore

$$\lim_{||\mathbf{f}||_2 \to +\infty} \frac{\partial \ell}{\partial \mathbf{w}_m} = 0. \tag{18}$$

For "incorrect" samples,

$$\lim_{||\mathbf{f}||_2 \to +\infty} \frac{\partial \ell}{\partial \mathbf{w}_m} = \begin{cases} +\infty & z_m = \max(z_1, ..., z_M) \text{ or } m = y; \\ 0 & \text{otherwise.} \end{cases} \tag{19}$$

Therefore, for a "correct" sample, whose feature has large norm, the network will failed to push the weights of other categories away from the feature, even if the weights are close to the feature. Also, for an "incorrect" sample, whose feature has large norm, the network will only consider the weights of the categories with largest logits and the weight of the ground-truth category of the sample.

Even if the other weights are close to the feature (but not close enough to be the weight of the maximum logits), the network will not push the weights away. Therefore, the weights vector may not be spread-out. Since the network will move the feature towards the corresponding weight in order to increase the logits and decrease the loss, if the weights are not spread-out, the features will not be spread-out.

## C  VISUALIZATION OF THE DISTRIBUTION OF THE FEATURE

The normalized histograms of the cosine similarity between the embeddings (learned with different models on the training set of the Car196 dataset) of samples of the same class (red) and samples of different class (blue) are shown in Fig. 4, for both the training (left) and test (right) sets. We show features trained with a total of 6 models: 1) Softmax, 2) BN (Softmax+BN), $\alpha = 4$, 3) BN, $\alpha = 8$, 4) BN, $\alpha = 16$, and 5) BN, $\alpha = 32$, and 6) HBN, $\alpha = 16 \rightarrow 4$,

A good embedding should exhibit the following properties:

- the features from the same category should be compact, which would transcribe as the red histogram being concentrated at the location where the cosine similarity is closed to 1;

- the features from different categories are spread-out, which would be observed as concentration of the blue histogram at the location where the cosine similarity is close to 0. In other words, two randomly sampled samples from different categories are close to orthogonal (Zhang et al., 2017).

- the embedding should be able to generalize to unseen classes of the same domain, i.e. the distribution on the test set should also exhibit compactness and spread-out properties.

Fig. 4(a) and 4(b) are the histograms for the model trained with the softmax function without any normalization. The main drawback of the feature is that the histogram of the feature of different categories is not concentrated (both for the training and test set), which means the samples from different categories are not spread-out (as detailed in Appendix B.2). Therefore the margin between different categories is not enough, or the features are not discriminative enough.

For BN models learned with different $\alpha$ values (Fig. 4(c)-4(j)), in training set (left hand side), we clearly see the trend that a smaller $\alpha$ value will lead to more compact features. Unfortunately, this trend doesn't hold for test set (right hand side). For the model trained with $\alpha = 4, 8$, although in the training set the histograms show nice compactness and spread-out properties, however the features are not compact at all on the test set. The reason is that, in metric learning problem, the training categories and the test categories are not overlapped, pushing the training samples of each category too close to each other can lead to overfitting.

We are especially interested in comparing the histograms of the features learned without heating-up (Fig. 4(g)-4(h)) and with heating-up (Fig. 4(k)-4(l)). Applying the proposed "heating-up" strategy, *i.e.* fine-tuning the network with smaller $\alpha$ and learning rate, makes the positive pairs more compact while keeping the negative pairs spread-out. It improves the clustering and retrieval performance over model trained without "heating-up" as shown in Sec. 5.3.

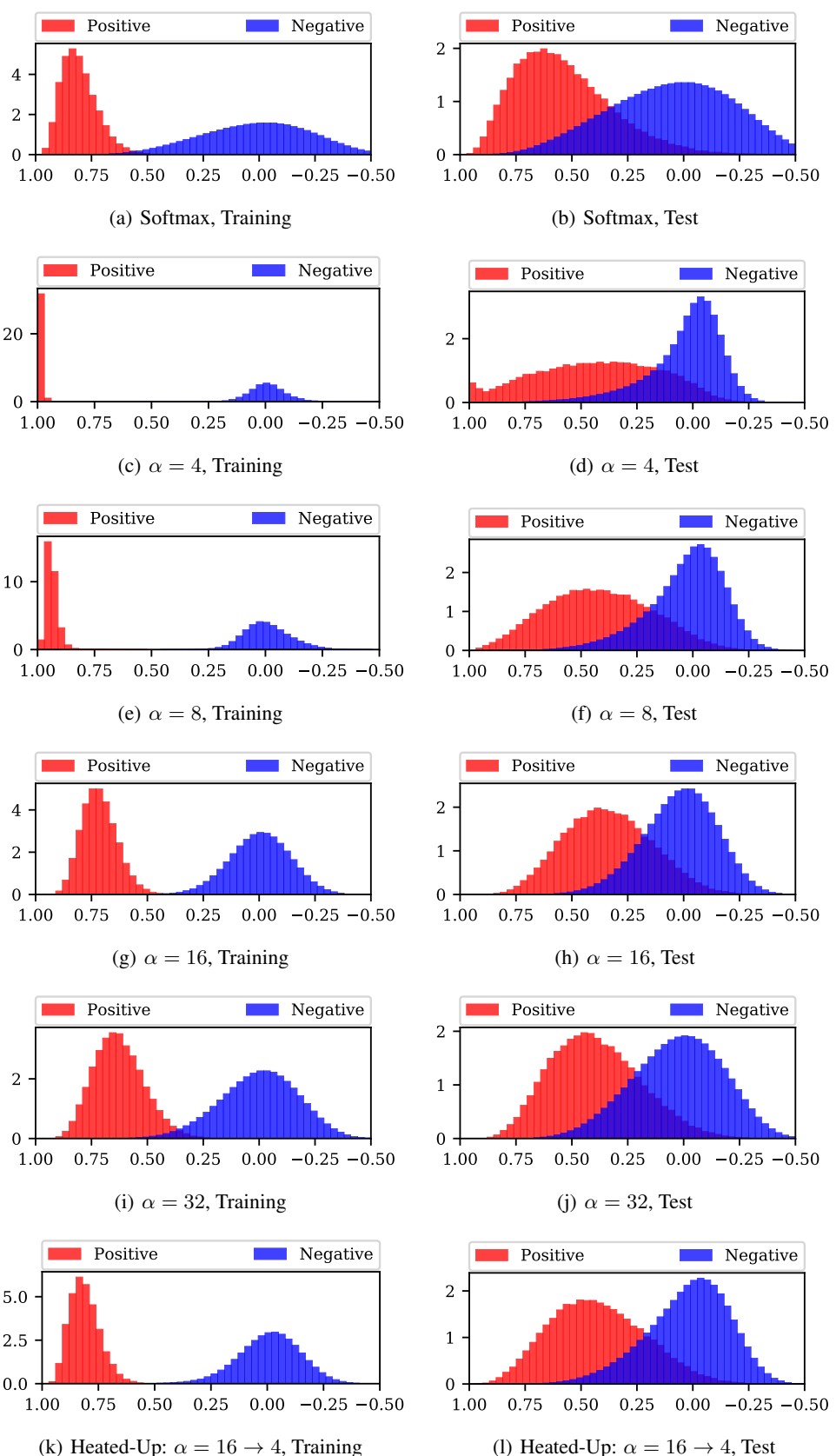

Figure 4: Embeddings trained with $\ell_2$-normalization and different $\alpha$ values on Car196 dataset.

## D  FINE-GRAINED CLASSIFICATION

Metric learning focuses on learning a compact and spread-out embedding. The goal of fine-grained classification focuses on learning a decision function to better classify the samples of different categories. Although the fine grained classification task doesn't require a compact and spread-out embedding, it's still interesting to see whether the proposed heating-up strategy helps fine-grained classification or not. We report in Table 4 the results of the fine-grained classification performance of different models on the Car196 and CUB200 datasets. The fine-grained classification model is trained with the predefined training/test split on all categories. Different from the training setting for metric learning in Sec. 5.3, here the training and test categories are the same. Except this, all other settings are the same. The classification accuracy over all categories is used for evaluation. The models trained with heating-up strategy outperforms the baseline model by a clear margin. The $\ell_2$ normalization model seems to work better than batch normalization model in fine-grained classification problem.

Table 4: Fine-grained Classification Performance for Car196 and CUB200 Datasets

|  | SM | LN | BN | HLN | HBN |
|---|---|---|---|---|---|
| CAR196 | 83.6 | 84.8 | 85.5 | **87.0** | **87.0** |
| CUB200 | 68.7 | 69.1 | 68.8 | **75.4** | 70.9 |

## E  COMPARISON WITH LEARNED $\alpha$

We compare the heating-up strategy with the strategy of directly learning the value of $\alpha$ proposed by Wang et al. (2017a). For learning $\alpha$, we apply different $\alpha$ values $[1, 2, 4, 8, 16, 32]$ as initialization. Other experiment settings are kept the same as in Sec. 5.

The initialized $\alpha$ value, final $\alpha$ value after convergence, NMI and Recall@1 are shown in Table 5. According to the result, training with a fixed intermediate $\alpha$ value (BN) outperforms the method with learnable $\alpha$. A similar conclusion is also reported by Ranjan et al. (2017). The final performances and the final $\alpha$ values are different with different initialized $\alpha$ values. The best performance is achieved when initializing $\alpha = 8$. It is interesting to observe that the final learned alpha values are always increased compared to the initial values, which is the opposite of what our heating-up strategy does. The reason can be interpreted by the properties mentioned in Appendix B. For "correct" samples, larger $\alpha$ value, which is equivalent to larger feature norm, results in smaller loss. Since most of the samples will become "correct" samples, the $\alpha$ value will become large.

We also apply the heating-up strategy to both learnable and fixed alpha settings. For learnable $\alpha$, the $\alpha$ values are reduced to $1/4$ of the final $\alpha$ values. The results show that the proposed heating-up strategy provides performance gains to all the settings. Learning with the fixed $\alpha$ value and heating-up (BN with heating-up, HBN) is still the winner overall.

Table 5: NMI and R@1(%) for Car196 with Learnable $\alpha$

|  | INITIAL $\alpha$ | FINAL $\alpha$ | NO HEATED-UP | | HEATED-UP | | |
|---|---|---|---|---|---|---|---|
|  |  |  | NMI | R@1 | HEAT-UP $\alpha$ | NMI | R@1 |
| LEARNED $\alpha$ | 1 | 15.6 | 62.48 | 67.92 | 3.9 | 64.73 | 69.96 |
|  | 2 | 21.2 | 62.15 | 67.86 | 5.3 | 66.61 | 72.67 |
|  | 4 | 28.9 | 63.04 | 68.80 | 7.2 | 66.29 | 73.03 |
|  | 8 | 37.0 | 63.87 | 70.83 | 9.3 | 67.38 | 74.11 |
|  | 16 | 41.2 | 62.96 | 69.04 | 10.3 | 67.33 | 73.80 |
|  | 32 | 44.7 | 63.11 | 69.80 | 11.2 | 65.17 | 72.37 |
| SM | N/A | N/A | 59.52 | 60.76 | N/A | N/A | N/A |
| BN | 16 | 16 | **65.81** | **71.12** | 4 | **68.10** | **74.70** |

## F  IMPORTANCE OF NORMALIZATION

Applying $\ell_2$ normalization to the classifier weights in network training helps to increase angular margin instead of Euclidean distance between different classes. It helps to get more discriminative feature representation.

In Table 6, we show the performances of the following models: 1) SM: Off-the-shelf structure with no normalization to feature or weights, 2) SM+$\alpha$: Off-the-shelf structure trained with different temperature values, 3) BN w/o WN: Trained with batch normalization to the features but no normalization to the weights, and 4) BN: Trained with both batch normalization to the features and $\ell_2$ normalization to the weights. BN w/o WN gets better performance than SM but worse performance than BN. For BN w/o WN models, the best performance is achieved when $\alpha = 16$, which is the same as BN. It shows that the temperature also works for models trained without weights normalization, and the weight normalization helps to improve the performance.

It shows that the temperature also works for models trained without weights normalization, and the weight normalization helps to improve the performance.

Table 6: NMI and R@1(%) for Car196 with Different Models

|  | SM | SM+$\alpha$ | | | BN w/o WN | | | BN | | |
|---|---|---|---|---|---|---|---|---|---|---|
|  |  | $\alpha = 8$ | $\alpha = 16$ | $\alpha = 32$ | $\alpha = 8$ | $\alpha = 16$ | $\alpha = 32$ | $\alpha = 8$ | $\alpha = 16$ | $\alpha = 32$ |
| NMI | 59.5 | 59.0 | 59.3 | 57.1 | 62.7 | 64.1 | 63.5 | 63.1 | 65.8 | 62.9 |
| R@1 | 60.8 | 64.1 | 63.8 | 60.5 | 67.5 | 68.9 | 68.8 | 68.7 | 71.1 | 69.5 |

## G  FINAL TRAINING ACCURACY WITH DIFFERENT $\alpha$ VALUES

As we mentioned in Sec.3, training with small $\alpha$ values will assign similar gradients to all "incorrect" samples and "correct" samples. However, since "incorrect" samples affect the accuracy the most, failing in giving larger gradients (i.e. higher priority) to those samples will affect the final accuracy. To verify this, we show the final training accuracy with different $\alpha$ values on Car196 dataset in Table 7. "1 - Train Acc" indicates the ratio of the number of "incorrect" sample and the number of total samples.

The result shows that, when training with small $\alpha$ value, especially when $\alpha = 2$, the training accuracy is significantly lower than the that of trained with larger $\alpha$ values. It verifies that training with large $\alpha$ values will focus more on the "incorrect" samples and provide higher training accuracy.

Table 7: Final Training Accuracy (%) for Car196 with Different $\alpha$ Values

|  | $\alpha = 2$ | $\alpha = 4$ | $\alpha = 8$ | $\alpha = 16$ | $\alpha = 32$ |
|---|---|---|---|---|---|
| TRAIN ACC | 88.68 | 98.77 | 99.59 | 99.57 | 99.60 |
| 1 - TRAIN ACC | 11.32 | 1.23 | 0.41 | 0.43 | 0.40 |

