# OpenReview forum: "Heated-Up Softmax Embedding"
_ICLR.cc/2019/Conference_

### Official Review · AnonReviewer2 · 2018-11-02
**It is a simple and interesting method, but lacks discussions and/or empirical evaluation in comparison with the prior work.**

**Rating:** 5
**Confidence:** 4

**Review:**

Summary:
This paper proposes a novel optimization strategy regarding softmax cross-entropy loss, to extract the effective features of well generalization in the framework of metric learning.
The authors focus on the "temperature" parameter in the softmax and through analyzing the role of the temperature in terms of gradient, propose the approach of heating-up softmax in which the temperature is varied from low to high in training.
And, the effects of normalization such as by l2 and BatchNorm are discussed in the framework of heated-up softmax.
The experimental results on metric learning tasks demonstrate the effectiveness of the proposed method in comparison with the other methods.

Comments:
Pros:
+ The idea of heating up the temperature in softmax is interesting, and seems novel in the literature of metric learning.
+ The performance improvement, especially produced by batchNorm-based normalization, is shown.

Cons:
- The formulation of tempered softmax with normalization is already presented in [Wang et al., 2017].
- The reason why the heating-up approach contributes to better metric learning is not clearly provided in a well convincing way.
- It lacks an important ablation study to fairly validate the method.
- The discussion/comparison is limited to the simple softmax function.

Although the reviewer likes the idea of heating up softmax, this paper can be judged as a borderline slightly leaning toward reject, due to the above weak points, the details of which are explained as follows.

- Formulation
The softmax equipped with temperature for the normalized features and weights are shown in [Wang et al., 2017]. The only difference from that work is the way to deal with temperature; in [Wang et al., 2017], the temperature is "optimized" as a trainable parameter, while it is dealt with in a hand-crafted way of heating up in this work. Honestly speaking, it is unclear which approach is better, though the optimization in [Wang et al., 2017] seems elegant as stated in that paper. The only way to validate this work compared to [Wang et al., 2017] is to empirically evaluate those two methods in the experiments. Such a comparison experiment is not found and it is a main flaw of this paper.

- Justification of the method
The gradients of the softmax cross-entropy loss parameterized with a temperature T are well analyzed in Sections 3.1&3.2. But, in Section 3.3, the reviewer cannot find the clear and convincing explanation for why the temperature T should be increased in the training. My question is: why don't you use alpha=4 consistently throughout the training?
 It might be related to the process of simulated annealing (though "temperature" is usually cooled down in SA), and more interestingly, it would also be possible to find connection with the work of [Guo et al., 2017]. In [Guo et al., 2017], the temperature in the softmax is optimized as a post processing for calibrating the classifier outputs. Though the calibration task itself is a little bit apart from the metric learning of the authors' interest, we can find in that paper an interesting result that the temperature is heated up to increase the confidence of the classifier outputs, which is quite similar to the process of fine-tuning by heating up softmax as done in this work. Therefore, the reviewer guesses that the effectiveness of heating up softmax can also be interpreted from the viewpoint of [Guo et al., 2017].

There is also less description about Figure 1; in particular, the reviewer cannot understand what Figure 1(d) means.

- Ablation study
To empirically resolve the above concerns, it is necessary to present the empirical comparison with the "static" softmax.
Namely, the methods of HLN/HBN should be carefully compared to LN/BN of "alpha=4", not only those of alpha=16 shown in Table 1&2; the comparison in Table 3 seems unfair since the authors apply the static softmax without normalization.
And, it would be better to show the performance of heated-up softmax "without" normalization to show the important role of the normalization, as done in [Wang et al., 2017].
In summary, since the proposed method is composed of a heating-up approach and feature normalization, the authors are required to validate the method from those two aspects, respectively, for increasing the significance of this paper.

- Other loss function
For achieving a compactness in feature representation, the simple softmax requires both temperature and normalization. It, however, is also conceivable to employ the other types of loss function for that purpose, such as [a] which is based on the (Mahalanobis) distance with taking into account the margin between categories. The distance based loss also embeds features into localized clusters, which satisfies the authors' objective in this work. To validate the proposed method, it is required to compare the method with such a different types of loss function.

[a] Wan, W., Zhong, Y., Li, T., & Chen, J. (2018). Rethinking Feature Distribution for Loss Functions in Image Classification, In CVPR2018, pp. 9117–9126.

---

> ### Author Response · Authors · 2018-11-26
> **Reply to R2**
>
> Q1: Missing comparison to [Wang et al., 2017], which learns the temperature.
> A1: We thank the reviewer for pointing out this recent related work. We added a comparison to [Wang et al., 2017] in the revised manuscript in the Appendix E. We applied different alpha values [1,2,4,8,16,32] as initialization. According to the results, training with a fixed intermediate alpha value (BN) outperforms the method with trainable alpha. A similar conclusion is also reported by Ranjan et al. (2017). It is interesting to observe that the final learned alpha values are always increased compared to the initial values, which is the opposite of what our heating-up strategy does. The reason for the lower performance is that learning the alpha value is equivalent to learning the norm of the feature. As mentioned in Appendix B, the classifier will tend to get larger alpha (larger feature norm) which results in a not compact feature.
>
> We also applied the heating-up strategy to both learnable and fixed alpha settings. The proposed heating-up strategy shows performance gains for all the settings. Learning with fixed alpha and the heating-up is still the overall winner.
>
> Q2: Provide a comparison of HLN/HBN to LN/BN for “alpha=4”
> A2: The comparison is given in Table 3. In the table, only the second column is the result of softmax without normalization. All other results are with normalization applied to both the feature and the weight which is the same setting as HBN. We’ve changed the statement there to make it clearer.
>
> Q3: Explain why the temperature T should be increased in the training and better explain Fig. 1(d).
> A3: Increasing temperature T will assign larger gradient to boundary samples which results in a compact feature that is beneficial for metric learning.
>
> The left-hand side of Fig. 1(d) shows the first step of “Heating-Up”, which at the beginning of the training, uses an intermediate alpha value to train the network. At the beginning, there are many “incorrect samples” and “boundary samples”, choosing an intermediate alpha value (red dashed line in Fig. 1(a)-(c)) will assign large gradients to the “incorrect samples”, which will quickly push them to be the “boundary samples”. The “boundary samples” will also be pushed towards the center.
>
> After the first step, the positions of the samples are shown in the right-hand side of Fig. 1(d). Continuing using an intermediate alpha will not give enough gradient for boundary samples. Therefore, we should use a smaller alpha (green dashed line in Fig. 1(a)-(c)) to assign larger gradient for updating the boundary samples, which will make the final embedding more compact.
>
> Q4: Show the importance of normalization.
> A4: Applying l2 normalization to the classifier weights in network training helps increase angular margin instead of Euclidean distance between different classes. It helps to get more discriminative feature representation. We’ve added Appendix F to explain this and provided experiments results of training the embedding without normalization.
>
> Q5: Discuss methods using different losses, such as [Wan et al. 2018].
> A5: We thank the reviewer for pointing out this reference. The reference proposes to use Mahalanobis distance instead of inner product in the softmax function and achieve good performance in image classification task. One main difference between the deep metric learning and [Wan et al. 2018] is that [Wan et al. 2018] requires the compactness and spread-out properties with Mahalanobis distance while in deep metric learning those properties are required in Euclidean space. Therefore, [Wan et al. 2018] may not give an optimal solution to the deep metric problem. Unfortunately, since the method is not originally designed for deep metric learning, given limited rebuttal time, we weren’t able to implement it for metric learning. We will definitely discuss this reference in our final version.

---

### Official Review · AnonReviewer1 · 2018-11-03
**Novelty**

**Rating:** 3
**Confidence:** 5

**Review:**

The introduction and the title does not match. Metric learning does not require to specify the dimension; while the embedding has to specify the reduced dimension. I feel confused that the authors mix these two concepts.

The objective in (1) is very close to that of t-SNE[5], where it uses the KL as the objective. Then other update formula are similar.

This paper facilitates the effect of temperature in the Softmax function to heuristically learn a compact and spread-out embedding. However, such an idea have been widely used and investigated in Reinforcement learning [1], Knowledge distillation [2], classification [3] and discrete variable optimization [4] and t-SNE visualization [5] etc. Thus, the insight about the temperature effect on the embedding from the second last layer, cannot be novel any more. Based on this, the proposed ``heating-up” strategy to leverage its effect on the embedding is heuristic, since the temperature parameter is manually set instead of automatically learning. In this case, I do expect the authors should provide more in-depth theoretical analysis.

The authors do not present more experimental results on the correlation between the final performance and this temperature setting.

Besides, as the alpha increases or decreases, the side-effect on the learning rate setting for the optimization have not clearly analyzed, which leaves more concerns on tuning performance.


[1] Sutton, R. S. and Barto A. G. Reinforcement Learning: An Introduction. The MIT Press, Cambridge, MA, 1998.
[2] Hinton G, Vinyals O, Dean J. Distilling the knowledge in a neural network. NIPS 2015.
[3] Guo, Chuan, et al. "On calibration of modern neural networks." ICML 2017.
[4] Jang E, Gu S, Poole B. Categorical reparameterization with gumbel-softmax. ICLR 2017.
[5] Maaten L, Hinton G. Visualizing data using t-SNE[J]. Journal of machine learning research, 2008, 9(Nov): 2579-2605.

---

> ### Author Response · Authors · 2018-11-26
> **Reply to R1**
>
> Q1: The proposed method is not Metric learning.
> A1: Conventional metric learning only learns a metric between different samples. However, Deep Metric Learning first learns a low-dimensional embedding for all the samples and use Euclidean distance on the new embedding as a metric. The embedding and the Euclidean distance are regarded as a whole as the ‘metric’. This terminology and setting have been widely used in recent years (Hoffer & Ailon, 2015, Harwood et al., 2017; Yuan et al., 2017; Wu et al., 2017; Song et al., 2016; 2017).
>
> Q2: The idea of using temperature and second to the last layer’s embedding is not novel.
> A2: We didn’t claim the idea of using temperature and second to the last layer’s embedding is the novelty of the paper. We have reviewed the related works on temperature and softmax embedding. As mentioned by R2 and R3, our novelty is to understand how different temperature values determine the distribution of the final embedding by assigning different gradients to different samples and the proposed heating-up strategy. The insight is completely different from existing literatures.
>
> Q3: The correlation between the final performance and temperature setting is not evaluated.
> A3: The correlation between the final performance and temperature is evaluated in Table 3. A new comparison of a learned alpha [Wang et al., 2017] versus our heating-up strategy is provided in Appendix E, showing that our proposed strategy significantly outperforms the learned approach.
>
> Q4: The side-effect on the learning rate setting is not analyzed.
> A4: The idea of heating-up is to use small alpha values to “fine-tune” the network. Choosing a learning rate that is 1/10 of the starting learning rate is a common strategy used for fine-tuning. Using a large learning rate will make the network converge to results similar to directly using a small alpha for training. The best learning rate can be chosen through cross-validation. However, 1/10 of the starting learning rate generally gave good performance for all datasets in our experiments.

---

### Official Review · AnonReviewer3 · 2018-11-05
**Heated-Up Softmax Embedding**

**Rating:** 8
**Confidence:** 4

**Review:**

This paper presents an interesting idea to improve the softmax embedding performance with heated-up strategy. It is well-written and the proposed method is easy to implement. Several experiments on metric learning datasets demonstrate the effectiveness of the proposed method.

The motivation to find a balance between the compactness and "spread-out" embedding is reasonable. The major weakness is the intermediate temperature selection, it might be a little tricky. How to generalize it to other applications?

The authors claim that "heated-up" strategy produces well generalized feature, but the rationale behind is unclear. And there is no quantitative analysis to support this point.

The starting temperature aims at pushing the “incorrect” samples to “boundary” samples and pushing the “boundary” samples to “centroid” samples. I would like to see the ratio of #incorrect/total and #boundary/total changed with different temperature in training process, i.e., alpha = 16, 4, 1. This experiment may help to verify the idea.

As mentioned in Section 3, multiple strategies could be defined to increase the temperature. It is interesting to design a multiple heat-up strategy. Does it help to improve the learning speed?

---

> ### Author Response · Authors · 2018-11-26
> **Reply to R3**
>
> Q1: How to select the intermediate temperature alpha.
> A1: A good intermediate temperature value can be selected by cross-validation. According to our experiment, choosing alpha value = 16 generally gives a good performance. According to the new experiment in Appendix E, we can learn an alpha value and apply the heating-up strategy based on that value.
>
> Q2: How "heated-up" strategy produces well generalized feature?
> A2: The explanation is given in Appendix C. For BN models learned with different alpha values (Fig. 4(c)-4(j)), in training set (left-hand side), we clearly see the trend that a smaller alpha value will lead to more compact features. Unfortunately, this trend doesn’t hold for test set (right hand side). For the model trained with alpha= 4,8, although in the training set the histograms show nice compactness and spread-out properties, the features are not compact at all on the test set. It’s a clear signal of overfitting.
>
> Comparing the histograms of the features learned without heating-up (Fig. 4(g)-4(h)) and with heating-up (Fig. 4(k)-4(l)), applying the proposed “heating-up” strategy, i.e. fine-tuning the network with smaller alpha and learning rate, makes the positive pairs more compact while keeping the negative pairs spread-out. We don’t observe clear overfitting phenomenon for the “heated-up” feature.
>
> Q3: How different alpha values change the ratio of #incorrect samples with #total samples and #boundary samples with #total samples?
> A3: To verify this, we show the final training accuracy with different alpha values on Car196 dataset in Table 7 in the revised manuscript. “1 - Train Acc” indicates the ratio of the number of “incorrect” sample and the number of total samples. The result shows that when training with small alpha value, especially when alpha = 2, the training accuracy is significantly lower (~11%) than the that of trained with larger alpha values (alpha=8, 16, 32). It verifies that training with large alpha values will focus more on the ``incorrect'' samples and provide higher training accuracy. To show the ratio of the #boundary samples to the #total samples is not really possible, since there is no formal definition of what a “boundary” sample is.
>
> Q4: Do multiple heating-up strategies improve learning speed?
> A4: We thank the reviewer for suggesting this good idea. This may help improve the learning speed. However, since the alpha value is only a hyperparameter, and the network needs sufficient time to respond to the change of the hyperparameter, combining different heating-up strategies may not help a lot.

---

### Meta-Review · Area_Chair1 · 2018-12-13
**promising quantitative results but limited contribution over previous work**

**Confidence:** 4
**Recommendation:** Reject

**Metareview:**

1. Describe the strengths of the paper.  As pointed out by the reviewers and based on your expert opinion.

- The method and justification are clear
- The quantitative results are promising.

2. Describe the weaknesses of the paper. As pointed out by the reviewers and based on your expert opinion. Be sure to indicate which weaknesses are seen as salient for the decision (i.e., potential critical flaws), as opposed to weaknesses that the authors can likely fix in a revision.

- The contribution is minor
- Analysis of the properties of the method is lacking.
The first point was the major factor in the final decision.

3. Discuss any major points of contention. As raised by the authors or reviewers in the discussion, and how these might have influenced the decision. If the authors provide a rebuttal to a potential reviewer concern, it’s a good idea to acknowledge this and note whether it influenced the final decision or not. This makes sure that author responses are addressed adequately.

Reviewer opinion was quite divergent but both AR1 and AR2 had concerns about the 2 weaknesses mentioned in the previous section (which remained after the author rebuttal).

4. If consensus was reached, say so. Otherwise, explain what the source of reviewer disagreement was and why the decision on the paper aligns with one set of reviewers or another.

No consensus was reached. The source of disagreement was on how to weigh the pros vs the cons. The final decision was aligned with the lower ratings. The AC agrees that the contribution is minor.